# LC-DAD–ESI-MS/MS and NMR Analysis of Conifer Wood Specialized Metabolites

**DOI:** 10.3390/cells11203332

**Published:** 2022-10-21

**Authors:** Andrzej Patyra, Marta Katarzyna Dudek, Anna Karolina Kiss

**Affiliations:** 1Department of Pharmacognosy and Molecular Basis of Phytotherapy, Medical University of Warsaw, 02-097 Warsaw, Poland; 2Doctoral School, Medical University of Warsaw, 02-091 Warsaw, Poland; 3Institut des Biomolécules Max Mousseron, Université de Montpellier, CNRS, ENSCM, 34293 Montpellier, France; 4Structural Studies Department, Centre of Molecular and Macromolecular Studies, Polish Academy of Sciences, 90-001 Łódź, Poland

**Keywords:** *Pinaceae*, conifers, liquid chromatography–mass spectrometry, chemical characterization, untargeted metabolomics, wood extracts

## Abstract

Many species from the *Pinaceae* family have been recognized as a rich source of lignans, flavonoids, and other polyphenolics. The great common occurrence of conifers in Europe, as well as their use in the wood industry, makes both plant material and industrial waste material easily accessible and inexpensive. This is a promising prognosis for both discovery of new active compounds as well as for finding new applications for wood and its industry waste products. This study aimed to analyze and phytochemically profile 13 wood extracts of the Pinaceae family species, endemic or introduced in Polish flora, using the LC-DAD–ESI-MS/MS method and compare their respective metabolite profiles. Branch wood methanolic extracts were phytochemically profiled. Lignans, stilbenes, flavonoids, diterpenes, procyanidins, and other compounds were detected, with a considerable variety of chemical content among distinct species. Norway spruce (*Picea abies* (L.) H.Karst.) branch wood was the most abundant source of stilbenes, European larch (*Larix decidua* Mill.) mostly contained flavonoids, while silver fir (*Abies alba* Mill.) was rich in lignans. Furthermore, 10 lignans were isolated from the studied material. Our findings confirm that wood industry waste materials, such as conifer branches, can be a potent source of different phytochemicals, with the plant matrix being relatively simple, facilitating future isolation of target compounds.

## 1. Introduction

The *Pinaceae* family encompasses 11 genera and about 230 species, making it the largest family of conifers as well as the *Gymnospermae*. Genera include *Abies* (47 spp.), *Cathaya* (1 sp.), *Cedrus* (3 spp.), *Keteleeria* (3 spp.), *Larix* (11 spp.), *Nothotsuga* (1 sp.), *Picea* (38 spp.), *Pinus* (113 spp.), *Pseudolarix* (1 sp.), *Pseudotsuga* (4 spp.) and *Tsuga* (9 spp.). The members of this family are monoecious, resinous trees distributed widely in the Northern Hemisphere, with just one equatorial crossover species [1]. Although trees from the *Pinaceae* family occupy an area which is 68.4% of all forests in Poland [2], species diversity is not very large, with only a few species: *Pinus sylvestris* L. (Scots pine), *Pinus mugo* Turra (mountain pine), *Pinus ×rhaetica* Brügger (Rhaetic pine) and *Pinus cembra* L. (Swiss pine), *Picea abies* L. (Norway spruce), *Larix decidua* Mill. (European larch) and *Larix polonica* Rac. (Polish larch), *Abies alba* Mill. (silver fir) and few introduced species, of which only *Pseudotsuga menziesii* Mirb. (Douglas fir), *Pinus strobus* L. (Weymouth pine), *Picea glauca* (Moench) Voss (white spruce), *Larix kaempferi* Lamb. (Japanese larch) and *Tsuga canadensis* Carrière (Canadian hemlock) can be found in significant numbers [3]. Most of the genera present in Poland are economically important for the timber industry, which is why they have been preferred in plantings since the 19th century [2].

With recent technological advances in the timber industry, logs and other wood products brought to sawmills are almost totally processed, leaving little to no waste. However, this is still not always the case. According to experts, the low technological level of a large part of Polish industrial plants, the lack of financial resources for proper waste management, the state’s unclear ecological policy, as well as a low level of public awareness are the reasons for the generation of substantial amounts of waste [4]. By-products from the timber industry, such as wood chips, sawdust, bark, and others, form 38.6% of the total wood flow in Europe [5]. Their high quality and solid structure could bring them many applications, yet they are mainly used to generate energy (as a biofuel) and in wood panel production [5,6]. As more and more emphasis is placed on limiting the increase in global average temperature (i.e., global warming), with the Paris Climate Agreement, and the European Union’s 2030 and 2050 climate strategies, a new, more responsible and efficient approach to wood by-products is well founded.

One of such wood by-products are wood knots, which are branch bases inside a stem. They are not particularly of interest to the timber industry, as they are considered defects of the wood (fibers in this portion of wood are perpendicular to the stem), they are much harder to chip than normal wood, and their presence in wood pulp may lead to the formation of paper sheets with inferior strength, light absorption, and surface properties [7]. Therefore, it is not surprising that knotwood is considered a by-product of many timber industry processes, with little applications or value to-date; most of it is burned to produce energy [7]. Knotwood, as well as branchwood (which contains many knots on its own), due to their high lignan content have great potential for industrial applications. Not only can we find much higher amounts of common dietary lignans—some of them with proven dietary anti-diabetic, anti-inflammatory, antioxidant and phytoestrogenic properties [8,9,10,11]—but also some much rarer compounds, that is, oligolignans [12], which have not yet been thoroughly studied. Only recently, a single lignan molecule 7-hydroxymatairesinol (HMRlignan) from *Picea abies* wood knots was registered as a dietary supplement [13]. Although many lignans, such as secolariciresinol or sesamin, have been previously found in high concentrations in oilseeds, 7-hydroxymatairesinol has been detected in the human diet quite recently, using high-resolution detection methods [14]. Both 7-hydroxymatairesinol and *Picea abies* wood knots extract were found to improve glucose metabolism, decrease insulin resistance, and indirectly inhibit adipogenesis [15].

Extracts from *Larix laricina* K. Koch inner bark have been used by the Cree people of Northern Quebec to treat symptoms related to type 2 diabetes mellitus, and is mentioned in their traditional pharmacopeia. Recently both in vitro [16,17] and in vivo experiments [18] have confirmed the bark’s anti-diabetic effects, namely hypoglycemic, antioxidant and increasing insulin sensitivity. Although the exact compound responsible for this action is not yet known, lignans such as lariciresinol and its derivatives have been isolated using bioassay-guided fractionation from this extract [19].

Many species from the *Pinaceae* family have been recognized as a rich source, not only of lignans, but also of flavonoids and other polyphenolics. Phenolic standardized extracts from the bark of different species, such as *Pinus pinaster* Aiton (Pycnogenol, Flavangenol, Oligopin), and *Abies alba* Mill. (Abigenol), have been thoroughly studied and implemented in pharmaceutical use [20,21,22,23].

The bark of conifer trees, another wood industry by-product, is considered one of the major sources of stilbenes—a rare group of polyphenols of which resveratrol is perhaps the most studied [24]. These metabolites are not only considered phytoalexins (due to their antimicrobial activities) but also possess various pharmacological properties, such as anti-diabetic, anti-inflammatory, antioxidant, and cardioprotective [25,26,27]. They are not limited to the bark, as they have been also found in the wood of pines and spruces [28,29].

Although there have been studies screening the composition of some conifer wood, there are none on the composition of endemic and introduced conifers in Poland. Due to potential differences caused by geographical location, studying the wood of most common Polish conifers may bring some insight into their polyphenolic composition as well as their activity and further possibility of use in the food or pharmaceutical industry.

This study aimed to analyze and phytochemically profile wood extracts of the *Pinaceae* family species, endemic or introduced in Polish flora, using the LC-DAD–ESI-MS/MS (liquid chromatography with diode array detector coupled with electrospray ionization tandem mass spectrometry) method and compare their respective metabolite profiles. We also aimed to extract some of the more uncommon compounds to elucidate their structure under NMR (nuclear magnetic resonance) and use them as reference compounds in MS/MS (tandem mass spectrometry) analysis. 

## 2. Materials and Methods

### 2.1. Chemicals and General Experimental Procedures

Methanol, chloroform, dichloromethane, ethyl acetate, formic acid, and n-hexane were purchased from POCh (Gliwice, Poland). Acetonitrile was purchased from Merck (Darmstadt, Germany). Water for HPLC experiments was prepared using the Milli-Q Plus system (Millipore, Billerica, MA, USA) (18.2 MΩ cm). All solvents used for chromatography were of gradient grade. Cyclolariciresinol, 7-hydroxylariciresinol (I), 7-hydroxylariciresinol (II), 7-hydroxymatairesinol, lariciresinol, matairesinol, nortrachelogenin, pinoresinol, secoisolariciresinol, and todolactol were isolated in our laboratory. Myricetin, dihydroquercetin (taxifolin), and epi-catechin were purchased from Serva (Heidelberg, Germany). Abietic acid and catechin were purchased from Sigma-Aldrich Chemie GmbH (Steinheim, Germany).

### 2.2. Plant Material and Extract Preparation

Branch wood of *Pinus cembra* L., *Pinus mugo* Turra, *Pinus strobus* L., *Pinus ×rhaetica* Brügger, *Abies alba* Mill., *Picea abies* L., *Picea glauca* (Moench) Voss, *Pseudotsuga menziesii* Mirb., *Tsuga canadensis* Carrière, and *Larix kaempferi* Lamb. was collected from the Polish Academy of Sciences Botanical Garden. Branch wood of *Larix decidua* Mill. and *Larix polonica* Rac. was collected from the University of Warsaw Botanical Garden. *Pinus sylvestris* L. branch wood was collected from Chojnów Landscape Park. The plant material was authenticated according to Flora Europaea by botanical gardens botanists. Voucher specimens have been deposited in the Plant Collection, Department of Pharmacognosy and Molecular Basis of Phytotherapy, Medical University of Warsaw, Poland. Plant material was dried at room temperature and shredded. Afterward, each sample was weighed (5.0 g), defatted with 50 mL *n*-hexane, and extracted with 50 mL of 90% (*v*/*v*) aqueous methanol under reflux for 2 h. The extract was then filtered through a 0.45 μm membrane, reduced in a rotary evaporator and lyophilized, resulting in dried methanolic extracts. 

### 2.3. Isolation of Lignans

Pure lignans were isolated from the branch wood of *Abies alba* Mill. (660 g) and *Pinus sylvestris* L. (250 g). Briefly, the plant material was degreased with 1 L of n-hexane and extracted 3 times in a SONIC-5 ultrasonic bath (POLSONIC, Poznań, Poland) with 100% methanol at 60 °C for 2 h and evaporated to dryness on a Rotavator rotary evaporator R-100 (Buchi, Flawil, Switzerland) yielding 43.7 g and 11.5 g of dry silver fir and Scots pine extracts, respectively, which was phytochemically characterized.

The crude extract of fir was subjected to silica gel column chromatography (65 × 5 cm) and eluted with a CHCL_3_-MeOH gradient (100:0 → 90:10) of 11 steps, 0.5 L each, to obtain 100 fractions of 55 mL, which were pooled into 10 main fractions (F1–F10) based on their TLC (60G F_254_ silica gel, mobile phase dichloromethane and methanol (93:7, *v*/*v*), reagent 1% vanillin solution in concentrated sulfuric acid and heat treatment at 105 °C) and LC-DAD–ESI-MS/MS profiles.

Fraction F6 (5.98 g) was rechromatographed on a silica gel column (40 × 3 cm) with CHCL_3_-MeOH gradient (95:5 → 85:15) of 10 steps, 250 mL each, to obtain 5 fractions (F6_1–F6_5) of 500 mL each. Using preparative HPLC, lignans were isolated from fraction F6_1 (90 mg): compounds **22** (4.5 mg) and **27** (21.4 mg), from fractions F6_2 (230 mg), F6_3 (2660 mg), F6_4 (440 mg) and F6_5 (480 mg): compounds **16** (49.4 mg), **20** (157.8 mg) and **23** (1238 mg), and from directly from fraction F8 (2610 mg): compounds **9** (15.7 mg), **12** (54.5 mg) and **20** (24.1 mg).

The crude extract of Scots pine was suspended in water and extracted 3 times with ethyl acetate. Ethyl acetate fraction (8.4 g) was then subjected to Sephadex LH-20 (Pharmacia, Stockholm, Sweden) chromatography (40 × 3 cm) and eluted with methanol to obtain 34 fractions of 8 mL, which were pooled into 5 main fractions (F1–F5) based on their TLC and HPLC profiles. From fraction F2 (110 mg), compounds **29** (9 mg), **30** (13 mg), and **32** (13 mg) were isolated using preparative HPLC and identified using NMR methods.

These compounds were used as reference compounds for ESI-MS/MS analysis.

### 2.4. Preparative Chromatography

Preparative HPLC was performed with a Shimadzu LC20-AP instrument (Shimadzu, Japan) using a Zorbax SB-C18 column (150 mm × 21.2 mm, 5 μm) (Agilent, Santa Clara, CA, USA) at a flow rate of 20.0 mL/min and detection at λ_1_ = 254 nm, λ_2_ = 280 nm. The mobile phase consisted of 0.1% formic acid in water (A) and 0.1% formic acid in acetonitrile (B) using the following gradient: 0–60 min, 15–100% B.

### 2.5. General NMR Procedures

^1^H, ^13^C, and 2D NMR spectra (ROESY, COSY, HSQC, HMBC) were obtained on a Bruker Avance III 500 NMR spectrometer (Bruker BioSpin, Rheinstetten, Germany), operating at 500 and 126 MHz, respectively, using standard pulse programs and 5 mm NMR tubes. All measurements were performed at 295 K. Spectra were recorded in methanol-d_4_ or in CDCl_3_ (Armar AG, Döttingen, Switzerland). In each case, spectra were calibrated at residual solvent resonances, at 3.31 ppm for ^1^H and 49.15 ppm for ^13^C (methanol-d_4_, compounds **9**, **12**, **16**, **20**, **22**, **23**, **27**, **30**, **32**) or 7.24 ppm for ^1^H and 77.23 ppm for ^13^C (CDCl_3_-d_6_, compound **29**).

### 2.6. Phytochemical Characterization by LC-DAD–ESI-MS/MS Method

LC-DAD–ESI-MS/MS analysis was performed on a UHPLC-3000 RS system (Dionex, Dreieich, Germany) with DAD detection (Dionex, Dreieich, Germany) and an AmaZon SL ion trap mass spectrometer with an ESI interface (Bruker Daltonik GmbH, Bremen, Germany). Separation was performed on a Zorbax SB-C18 column (150 mm × 2.1 mm, 1.9 μm) (Agilent, Santa Clara, CA, USA). The mobile phase consisted of 0.1% formic acid in water (A) and 0.1% formic acid in acetonitrile (B) using the following gradient: 0–60 min, 15–100% B, then 10 min of equilibration. Samples for LC-DAD–ESI-MS/MS analysis were prepared by dissolving dried extracts in 0.1% formic acid in methanol at the concentration of 10 mg/mL. Standards were prepared in the same way at the concentration of 1 mg/mL). The flow rate was 0.2 mL/min, injection volume was 5 μL, column temperature was set at 25 °C. The LC eluate was introduced into the ESI interface without splitting, and compounds were analyzed in both positive and negative ion mode with the following settings: nebulizer pressure of 40 psi, drying gas flow rate of 9 L/min, nitrogen gas temperature of 300 °C, and a capillary voltage of 4.5 kV. The mass scan ranged from 100 to 2200 m/z. Low-energy collision-induced dissociation (less than 100 eV) was used to obtain MS/MS spectra, with collision energies chosen automatically based on precursor ion masses by the integrated SmartFrag mode.

UV spectra were recorded in the range 190–400 nm. Compounds were identified by comparing their retention time and UV-visible and mass spectra with those obtained from reference compounds and/or tentatively identified by comparison with literature information.

## 3. Results

Through comprehensive LC-DAD–ESI-MS/MS analysis of *Pinus sylvestris* L., *Pinus cembra* L., *Pinus mugo* Turra, *Pinus strobus* L., *Pinus ×rhaetica* Brügger, *Abies alba* Mill., *Picea abies* L., *Picea glauca* (Moench) Voss, *Pseudotsuga menziesii* Mirb., *Tsuga canadensis* Carrière, *Larix decidua* Mill., *Larix polonica* Rac., and *Larix kaempferi* Lamb. branch wood methanolic extracts, 40 compounds were identified or partially identified based on the elution order, UV maxima, pseudomolecular and fragmentation ions in both positive and negative ion modes and comparison to literature data (Table 1). Presence of some flavan-3-ols, flavonoids and abietic acid was confirmed by comparison with commercial standards. In the case of lignans, most of the compounds were isolated and their structures were confirmed by NMR analysis. Then, their presence in selected wood extracts was confirmed by LC-DAD–ESI-MS/MS analysis by comparison with pure compounds. 

The investigated plant material contained specialized metabolites belonging to flavan-3-ols, flavonoids, lignans, sesquilignans, stilbenes, sesquiterpenes, and diterpenes. The presence of identified compounds in each studied species is presented in (Table 2). The UV chromatograms of each wood extract are provided in Appendix A. 

### 3.1. Flavan-3-ols

Eight compounds belonging to the group of flavan-3-ols were identified in selected wood extracts. Compounds **5** (t_r_ = 4.5 min) and compound **7** (t_r_ = 6.6 min) exhibited a pseudomolecular ion at m/z 289 and had identical fragmentation patterns. Based on their elution order and comparison to reference standards, these compounds were identified as catechin and epi-catechin, respectively. Whilst catechin was present in all studied plant samples, we were not able to detect epi-catechin in *P. sylvestris*, *P. mugo*, *P. ×rhaetica*, and both spruces (Table 2).

Four compounds, **2** (t_r_ = 3.3 min), **3** (t_r_ = 3.6 min), **6** (t_r_ = 5.2 min) and **15** (t_r_ = 11.6 min), exhibited a pseudomolecular [M−H]^−^ ion at m/z 577, which fragmented into m/z 559 [M−H−18]—loss of water, m/z 451 [M−H−126]^−^ from heterocyclic ring fissure fragment, m/z 407 [M−H−170]^−^—product of retro-Diels–Alder reaction followed by the loss of water and m/z 289, which corresponds to (epi)catechin. Based on fragmentation patterns, they were all characterized as type B dimeric procyanidins [41]. Previously, procyanidins B1-B4 have been isolated from the bark of *Larix gmelinii* (Rupr.) Kuzen. [42] and *Pseudotsuga menziesii* Mirb. [31]. In each plant sample, at least two dimeric procyanidins B were present. Larch wood was the richest in procyanidins, containing all four compounds in comparison to all studied genera (Table 2).

Additionally, one more procyanidin was detected in all wood extracts—compound **4** (t_r_ = 4.2 min). The main pseudomolecular ion was at m/z 865 and fragmented into m/z 695, m/z 577, m/z 407 and m/z 289. Based on the fragmentation pattern [43], this compound was assigned as type B trimeric procyanidin (procyanidin C).

Furthermore, compound **1** (t_r_ = 3.0 min) gave a pseudomolecular ion at m/z 305, which fragmented into m/z 219, m/z 179 and m/z 137. This fragmentation pattern was typical for gallocatechin [41]. The presence of this flavan-3-ol were found in *A. alba*, all studies *Pinus* spp., and *Larix* spp. (Table 2).

### 3.2. Flavonoids

A total of 10 flavonoids were found in studied wood extracts. Most of them were present in aglycone form, with just one glycoside form detected. Compound **18** (t_r_ = 12.4 min) was the most abundant flavonoid in these extracts, as it was not present only in *A. alba* and *T. canadensis* (Table 2), and had the highest peak of all flavonoid aglycones. Compound **18** had its UV maximum at 280 nm and exhibited a pseudomolecular ion at m/z 303, which fragmented into m/z 285 and m/z 177. The compound was then identified by comparison with the reference standard and comparison to the literature data [44,45,46] as taxifolin (syn. dihydroquercetin). Another major flavonoid compound **13** (t_r_ = 10.2 min) had its UV maximum at 287 nm and its pseudomolecular ion at m/z 465 and gave a primary fragment ion at m/z 303 [M−H−162]^−^ corresponding to the cleavage of hexose (probably glucose). Thus, compound **13** was assigned as taxifolin hexoside, which was consistent with the fragmentation pattern reported previously [44].

Four more flavanonols were detected in *Pinaceae* wood, with their UV maxima at around 290 nm. Compound **8** (t_r_ = 7.7 min) exhibited a pseudomolecular ion at m/z 319 and fragment ions at m/z 301 [M−H−18]^−^ corresponding to the loss of water, at m/z 193 and m/z 125. By comparing fragmentation pattern with literature data [47], compound **8** was assigned as dihydromyricetin (syn. ampelopsin, ampeloptin). It was found in samples from *P. mugo*, *P. rhaetica*, *L. decidua*, and *L. polonica* (Table 2). Compound **11** (t_r_ = 8.4 min) had its pseudomolecular ion at m/z 287, which fragmented into m/z 259 [M−H−28]^−^, corresponding to the loss of CO, m/z 243 [M−H−44]^−^, corresponding to the loss of CO_2_ and m/z 181. The fragmentation pattern matched to the literature data of dihydrokaempferol (syn. aromadendrin) [47]. Dihydrokaempferol was only found in two larch species, namely European and Polish larch (Table 2). Compound **33** (t_r_ = 22.0 min) exhibited a pseudomolecular ion at m/z 271 and fragment ion at m/z 253 [M−H−18]^−^ corresponding to the loss of water. It was tentatively identified, comparing to literature data [48], as pinobanksin, and compound **36** (t_r_ = 30.5 min), which exhibited a similar fragmentation pattern, was assigned as pinobanksin 3-*O*-acetate, also consistent with the data found in the literature [49]. The presence of pinobanksin was not genus-specific, while pinobanksin 3-*O*-acetate was only found in *P. strobus* wood (Table 2).

We have found presence of two flavanones, compound **24** (t_r_ = 15.4 min) and **35** (t_r_ = 30.3 min), which had their UV maxima at around 290 nm. Compound **24** presented a similar fragmentation pattern to that of compound **11**. Based on the elution profile and referring to the literature [50], it was tentatively assigned as eriodictyol. It was present in all larch species, Douglas fir, mountain pine, Swiss pine, and Weymouth pine. Pseudomolecular ion of compounds **35** was at m/z 255 and fragmented into m/z 213 and m/z 211. By comparison with the literature data [51] it was assigned as pinocembrin. We have found its presence in most pine species (apart from *P. cembra*) and in Douglas fir (Table 2).

We report presence of two flavonols in studied wood extracts, which were characterized by their UV maxima at around 370 nm. Compounds **25** (t_r_ = 15.7 min) and **31** (t_r_ = 19.8 min) had typical fragmentation patterns of myricetin and quercetin, respectively [52], which was confirmed by comparison with reference standards. Quercetin and myricetin were found in larches and Douglas fir. Additionally, Weymouth pine contained quercetin, while myricetin was also present in mountain pine (Table 2).

### 3.3. Lignans

Lignans have been identified as major constituents in fir and pine species. However, the spectral data for this class of compound is rather limited, thus, MS/MS analysis of this group of plant metabolites had to be supported by the isolation of studied compounds and elucidation of their structures by NMR. Ten lignans have been isolated using chromatographic methods, namely 7-hydroxylariciresinol (I) (**9**), todolactol (**12**), 7-hydroxylariciresinol (II) (**16**), cyclolariciresinol (**20**), 7-hydroxymatairesinol (**22**), secoisolariciresinol (**23**), lariciresinol (**27**), nortrachelogenin (**29**), pinoresinol (**30**) and matairesinol (**32**), with their structures confirmed through NMR analysis and comparison with literature data [53,54,55,56]. Isolated compounds were then used as reference standards in phytochemical profiling of conifer wood extracts. 

The 1D NMR and ESI-MS/MS spectra along with NMR chemical shifts of all isolated item are presented in Appendix A. 

Based on the LC-DAD–ESI-MS/MS analysis, three compounds, **9** (t_r_ = 7.8 min), **12** (t_r_ = 9.9min), and **16** (t_r_ = 11.6 min), had their pseudomolecular ion at m/z 375 [M−H]^−^ and UV maxima at around 225 and 280 nm. There have been previously reported three lignan isomers in conifer wood, that could give pseudomolecular ion at m/z 375: liovil, todolactol and 7-hydroxylariciresinol [57]. Unfortunately, no ESI-MS/MS fragmentation pattern data could be found for liovil and 7-hydrolariciresinol., with just one source for todolactol [57]. Through NMR analysis, we were able to assign compounds **9** and **16** as 7-hydroxylariciresinol diastereoisomers, respectively. Fragmentation of m/z 375 ion corresponding to compound **12** gave m/z 327, m/z 191, and m/z 176 which was consistent with the fragmentation pattern proposed for todolactol [57], and in agreement with fragmentation of our isolated standard. The occurrence of lignans in different conifer species was as follows: 7-hydroxylariciresinol (I) was found in silver fir and both spruces, todolactol was additionally present in European and Japanese larch, Douglas fir and Canadian hemlock. The lignan 7-hydroxylariciresinol (II) was present in at least one species of each genus except spruces (Table 2). LC-DAD–ESI-MS/MS analysis did not detect the signal that could correspond to liovil.

Compounds **30** (t_r_ = 19.1 min) and **32** (t_r_ = 21.6 min) exhibited pseudomolecular ions at m/z 357 [M−H]^−^ and similar fragmentation patterns. Compound **32** gave a characteristic primary product ion for dibenzylbutyrolactone lignans [M−H−44]^−^ at m/z 313, which corresponds to the loss of CO_2_ from the lactone ring [57], and that was not observed for compound **30**. Compound **30** produced a fragment ion at m/z 342 [M−H−15]^−^ corresponding to the loss of methyl radical, and at m/z 151 (guaiacyl) and m/z 136, which were the products of α,β-cleavage in the side chain. This fragmentation pattern was characteristic for furofuran lignans [57]. According to the fragmentation, elution order in RP-LC, and comparison with reference standards, compound **30** was identified as pinoresinol, while compound **32** was identified as matairesinol. Both compounds were also isolated, and their structure was confirmed through NMR analysis. Pinoresinol was only present in Scots pine wood and matairesinol could be found in nearly every plant material except silver fir and Douglas fir (Table 2).

Compounds **22** (t_r_ = 15.0 min) and **29** (t_r_ = 18.3 min) exhibited their pseudomolecular ions at m/z 373 [M−H]^−^. They both formed a fragment ion at m/z 355 [M−H−18]^−^ corresponding to the loss of water. For compound **22,** a fragment ion at m/z 311 [M−H−62]^−^ could be found, corresponding to the loss of water and CO_2_ from the lactone ring (again suggesting the compound belonged to the dibenzylbutyrolactone group). On the other hand, compound **29** did not produce a fragment ion at m/z 311; instead, an ion at m/z 327 [M−H−46]^−^ which corresponds to the loss of water and CO was observed,. Both fragmentation patterns and differences among them have been previously observed [58]. According to the fragmentation, elution order in RP-LC, and comparison with reference standards, compound **22** was identified as 7-hydroxymatairesinol and compound **29** was identified as nortrachelogenin (syn. 8′-hydroxymatairesinol). They were also isolated and their structure was confirmed using NMR methods. Nortrachelogenin and 7-hydroxymatairesinol were observed in most analyzed conifer wood, only excluding mountain and Rhaetic pine for the former, and additionally excluding Scots pine for the latter. Compound **21** (t_r_ = 14.0 min) had a pseudomolecular ion at m/z 419 [M−H+HCOOH]^−^ which fragmented into m/z 373 [M−H]^−^ corresponding to the loss of formate adduct. Based on literature data, it was tentatively identified as α-conidendric acid [57]. It was found in silver fir, mountain pine, Weymouth pine, all larch species, and Canadian hemlock (Table 2).

Compound **27** (t_r_ = 16.1 min) had its pseudomolecular ion at m/z 405 which fragmented to m/z 359 [M−H]^−^, corresponding to the presence of formyl adduct. We also observed another compound, **20** (t_r_ = 12.7 min), exhibiting its pseudomolecular ion at m/z 359 [M−H]^−^. Differences in further fragmentation could be observed for these compounds. For peak **27** we could observe a fragment ion at m/z 329 [M−H−30]^−^ resulting from the loss of formaldehyde. Compound **20** exhibited fragment ions at m/z 344 [M−H−15]^−^, corresponding to the loss of methyl radical, at m/z 313 [M−H−46]^−^, formed by the loss of methoxyl and methyl radicals, and at m/z 189, which was a product of β-cleavage in the side chain. Both fragmentation patterns have been previously observed [57] for lariciresinol (**27**) and cyclolariciresinol (syn. isolariciresinol) (**20**), respectively, which was then confirmed by isolating these compounds and elucidating their structure through NMR analysis. Cyclolariciresinol was only found in silver fir, while lariciresinol was present in all larch species, silver fir, Douglas fir, Weymouth pine, Canadian hemlock, and white spruce (Table 2).

Apart from cyclolariciresinol (**20**), one other butanediol lignan has been found. For compound **23** (t_r_ = 15.3 min) a pseudomolecular ion could be observed at m/z 361 [M−H]^−^ and fragment ions at m/z 346 [M−H−15]^−^, corresponding to the loss of methyl radical, at m/z 313 [M−H−48]^−^, formed by the loss of formaldehyde and water in the diol structure and at m/z 299 [M−H−48−15]^−^, corresponding to both rearrangements. This structure was compared to fragmentation data described in the literature [57], isolated, and identified as secoisolariciresinol. This was the dominant metabolite of silver fir and was also found in all larch and spruce species, mountain and Weymouth pine, Douglas fir, and Canadian hemlock (Table 2).

### 3.4. Sesquilignans

Two sesquilignans were found in conifer wood. Compound **26** (t_r_ = 15.7 min) exhibited a pseudomolecular ion at m/z 557 [M−H]^−^ and fragment ions at m/z 539 [M−H−18]^−^ corresponding to the loss of water, m/z 525 [M−H−32]^−^, m/z 521 [M−H−36]^−^, m/z 509 [M−H−48]^−^, formed by the loss of formaldehyde and water in the diol structure, m/z 415 and m/z 361 [M−H−196]^−^, corresponding to the cleavage of guaiacylglyceryl moiety. The fragmentation pattern was in accordance with the data for oligolignans isolated from Norway spruce and Scots pine knots [12]. Based on fragmentation, this compound was tentatively identified as secoisolariciresinol 4-*O*-guaiacylglyceryl ether. It was found in silver and Douglas fir, as well as all larch species and mountain pine (Table 2).

Compound **28** (t_r_ = 16.7 min) had its pseudomolecular ion at m/z 555 [M−H]^−^ and fragment ions at m/z 525 [M−H−20]^−^, m/z 507 [M−H−48]^−^ from the loss of formaldehyde and water, m/z 359 [M−H−196]^−^, corresponding to the cleavage of guaiacylglyceryl moiety, and at m/z 329 [M−H−196−30]^−^, resulting from the loss of formaldehyde. This compound was tentatively identified as lariciresinol 4-*O*-guaiacylglyceryl ether in accordance to previous MS/MS data [12]. It was only found in silver fir wood (Table 2).

### 3.5. Stilbenes

The Pinaceae family, is known to be a rich source of stilbenes [28,29,41]. In our study we found six stilbenes in conifer wood, mostly in spruce and pine species. They had characteristic UV maxima either at around 320 nm (glycosidic form) or 300 nm (aglycone form). Compounds **10** (t_r_ = 8.1 min) and **14** (t_r_ = 10.8 min) exhibited two pseudomolecular ions at m/z 811 [2M−H]^−^ and m/z 405 [M−H]^−^ with similar fragmentation patterns. Primary ion at m/z 405 fragmented into m/z 243 [M−H−162]^−^, corresponding to the cleavage of hexose, and showed fragments at m/z 225 [M−H−18]^−^, corresponding to the loss of water, m/z 201 [M−H−42]^−^, corresponding to the losses of C_2_H_2_O, characteristic for stilbenoids, m/z 173 [M−H−42−28]^−^, and m/z 159 [M−H−42−42]^−^. Based on elution order and comparison with literature data [28,59,60,61], compounds **10** and **14** were identified as *trans*-astringin (piceatannol 3-*O*-glucoside) and *cis*-astringin, respectively. *Trans*-astringin was only observed in spruce species, while *cis*-astringin was also present in larch species (Table 2).

Compounds **17** (t_r_ = 11.7 min) and **19** (t_r_ = 12.5 min) were only found in spruces. The main pseudomolecular ion for compound **17** was at m/z 389 [M−H]^−^ and gave one fragment ion at m/z 227 [M−H−162]^−^, corresponding to the cleavage of hexose, and showed fragments of aglycon at m/z 185 [M−H−42]^−^ and 183 [M−H−42−H_2_]−, corresponding to the losses of C_2_H_2_O, characteristic for stilbenoids, m/z 157 [M−H−42−28]− and m/z 143 [M−H−42−42]^−^ [61]. It was tentatively identified as piceid (resveratrol 3-*O*-glucoside) in comparison with literature data [60]. Compound **19** exhibited a pseudomolecular ion at m/z 465 [M−H+HCOOH]^−^ which showed fragment ions at m/z 419 [M−H]^−^, corresponding to the loss of formyl adduct, at m/z 257 [M−H−162]^−^, corresponding to the cleavage of hexose, at m/z 242 [M−H−162−15]^−^ and at m/z 241 [M−H−162−16]^−^ [28]. It was tentatively identified as isorhapontin (isorhapontigenin 3-*O*-glucoside).

Compounds **34** (t_r_ = 26.3 min) and **37** (t_r_ = 33.6 min) were characterized as stilbenes based on their primary pseudomolecular ion and UV maxima at around 300 nm. Comparing mass of [M−H]^−^ ion at m/z 211 [M−H−]^−^ and m/z at 225 [M−H]^−^ with literature reports on stilbenes isolated previously from conifer wood and their elution order [53], these compounds were tentatively identified: **34** as pinosylvin and **37** as pinosylvin monomethyl. Moreover, compared to MS/MS data from the literature, pinosylvin (**34**) demonstrated a fragment ion at 169 m/z [M−H−42]^−^ corresponding to the losses of C_2_H_2_O, characteristic for stilbenoids [59,62]. Pinosylvin monomethyl ether (**37**) pseudomolecular ion gave a fragment ion at 210 m/z [M−H−15]^−^, corresponding to the loss of methyl radical. Their occurrence was limited to the genus *Pinus*, with only Scots pine and mountain pine containing all the above (Table 2). 

### 3.6. Sesquiterpenoids and Diterpenoids

We were able to identify one sesquiterpenoid and two diterpenes in conifer wood— all of them in positive ion mode. Compound **38** (t_r_ = 41.0 min) exhibited a pseudomolecular ion at m/z 265 [M+H]^+^ and fragmented into m/z 251, m/z 233, m/z 205, m/z 187 m/z 176 and m/z 83. It was tentatively identified as dehydrojuvabione by comparison with MS/MS data found in literature [63]. Compounds **39** (t_r_ = 52.0 min) and **40** (t_r_ = 52.3 min) both exhibited a pseudomolecular ion at m/z 303 [M+H]^+^ with similar fragmentation patterns: m/z 285, m/z 257 and m/z 123. Based on UV maxima, 251 nm for compound **39,** and 241 nm for compound **40**, their elution order, comparing this data to the literature [64,65] and the reference standard, we were able to identify compound **39** as neoabietic acid and compound **40** as abietic acid. Dehydrojuvabione was only found in *A. alba* while both abietane-type resin acids were present in all studied conifer species wood (Table 2).

## 4. Discussion

Conifer wood and waste materials from its processing in the paper and wood industry (such as knots, bark and branches) have been identified as a good source of many polyphenols. Willfor and his team characterized wood knots of spruces, pines and firs, isolating many lignans in the process [34,39,66,67]. More recently, Gabaston et al. not only characterized wood knots of *Pinus pinaster* Aiton using LC-MS/MS, but also studied their antifungal activity against grapevine pathogens [53]. Another fine work offered LC-MS/MS analysis of stilbenes in *Picea jezoensis* Carrière bark [28]. To the best of our knowledge, most of the wood studied here was never characterized before using LC-MS/MS techniques. This study was also the first attempt at phytochemical profiling of wood from *P. ×rhaetica* and *L. polonica*, as well as the first to compare the composition of *Pinaceae* species growing in Poland.

Referring to previous research, our study confirms that conifers from the *Pinaceae* family are rich in otherwise scarce compounds, such as lignans and stilbenes, offering a large variety of these structures.

Lignans are a group of natural polyphenols (and one of the most lipophilic), located in plant cell walls. According to the recent nomenclature: lignans dimers of two coniferyl, sinapyl, 4-hydroxycinnamyl alcohol or similar alcohol monomers. Some authors restrict the term lignan only to those molecules coupled by the central carbon of the sidechain (i.e., 8,8′ or β, β’ dimers) [68]. They were first identified in conifer wood, which is why they often obtain their names from those species, e.g., pinoresinol from *Pinus nigra* Aiton and lariciresinol from *Larix decidua* Mill. [69]. Although lignans have been detected in numerous plants, with many being part of human diet, their content is usually very low. Apart from conifer wood knots (where they can accumulate in amounts up to 30% (w/w)), flaxseed and sesame seeds are considered rich but challenging sources of lignans. In this context, studied wood from the *Pinaceae* family offers a much easier-to-extract source of these polyphenols, with much fewer steps required to obtain pure compounds (directly in aglycone form with much larger structure variety). Wood from all studied species contained detectable amounts of lignans, with silver fir being the richest source of these polyphenols. Although *A. alba* offered large quantities of lignans, especially secoisolariciresinol and lariciresinol, *P. sylvestris* was far richer than all other species in nortrachelogenin, pinoresinol, and matairesinol. This is particularly interesting, as Scots pine is the most abundant and economically most important tree in Europe. Another important notion from our study is the detection of a lignan (namely matairesinol) in *P. cembra*, which was previously described as the only pine not containing lignans [7].

Through our analysis, we did not detect few lignans that have been previously found in some of the studied species. Willfor et al. reported presence of matairesinol, pinoresinol, and nortrachelogein in *A. alba* wood knots [34], but we did not observe any signal at expected retention times. In the same study, presence of secoisolariciresinol monomethyl ether and dimethyl ether was established. We cannot rule out the presence of these compounds in our samples, as pseudomolecular ions [M−H]− with expected mass for these lignans were observed. Unfortunately, no fragmentation pattern for them could be found in the literature, nor could we isolate them from the plant material. Similar problems occurred with analysis of sesquilignans, for which only few fragmentation patterns could be found, thus limiting our ability to identify them through simple LC-MS/MS analysis. Noticing that many lignans and sesquilignans did not have ESI fragmentation patterns reported previously, our study fills this knowledge gap for at least some of these compounds.

Species from the *Pinaceae* family are considered as one of the richest in stilbenes content. These rare polyphenols consist of two aromatic rings linked by a C_2_ ethylene sidechain, formed through cyclization of 3 malonyl-CoA with cinnamoyl-CoA, coumaroyl-CoA or caffeoyl-CoA, followed by hydroxylation of aromatic rings [70]. It has been previously reported that stilbenes are abundant constituents of *P. abies* and *P. jezoensis* bark, with their levels in wood being much lower [28,32]. Hovelstad et al. reported no presence of stilbenes in *P. abies* heartwood and knotwood; however, their analysis was limited to pinosylvins [29]. Our study offered analysis of stilbenes from different groups and included species not screened for stilbenes before. We have shown presence of stilbenes in all studied pines, spruces, and larches wood extracts, though their structure and nature varied among species. Stilbenes present in pine wood belonged to the pinosylvins, which were more lipophilic than those found in spruce and larch wood, which mostly occurred in glycosidic form. In the case of spruces, the UV signal for *trans*-astringin was much stronger than that of most abundant lignan reported in these species, i.e., 7-hydroxymatairesinol, which could imply higher content of stilbenes than lignans in spruce wood.

As was the case with lignans, fragmentation patterns for stilbenes are rarely reported in the literature, thus making LC-DAD–ESI-MS/MS analysis of these compounds difficult. This study offers MS^2^ fragmentations for some of these compounds, yet we see the need for further exploration of ESI fragmentation. Additional focus should be directed at differentiation of cis/trans isomers and the site of glucose substitution. In the case of our study, no differences were observed for such compounds and identification had to be based on elution order.

Our analysis also offered an insight into flavonoid, flavan-3-ol, and diterpene composition of conifer wood. These compounds were present in all studied samples, with differences concerning structures and their exact content. Larches and Douglas fir could be considered the richest in flavonoids and flavan-3-ols.

## 5. Conclusions

Overall, results obtained from this study show that conifer wood is a significant source of different phytochemicals, with many rare structures present. Silver fir wood could be a rich source of lignans, pine wood offer both lignans and stilbenes aglycones, while spruce wood had stilbenes glycosides as dominant metabolites. Larch and Douglas fir wood can be considered abundant sources of flavonoids and flavan-3-ols.

Furthermore, isolation of these metabolites from the plant matrix can be easily achieved. As these plant metabolites have already proven numerous pharmacological and dietary activities, they may find applications in pharmaceutical, cosmetic and food industries. Additionally, LC-ESI-MS has been established as quite a successful method for conifer wood metabolites detection and identification, allowing for simultaneous identification of phytoconstituents from different groups. 

## Figures and Tables

**Table 1 cells-11-03332-t001:** Retention time, UV, and ESI-MS/MS data of the compounds identified in *Pinaceae* species wood using LC-DAD–ESI-MS/MS analysis.

Compounds	Retention Time [min]	UV_max_ [nm]	ESI ^1^	MS1 [m/z]	MS2 [m/z]	Group	Presence in *Pinaceae* spp.
**1.** **gallocatechin**	3.0	199, 277	[M−H]^−^	305	219, 179, 137	flavan-3-ol	[30]
**2.** **dimeric procyanidin B (I)**	3.3	230, 282	[M−H]^−^	577	559, 451, 407, 289	flavan-3-ol	[31]
**3.** **dimeric procyanidin B (II)**	3.6	207, 238, 280	[M−H]^−^	577	559, 451, 407, 289	flavan-3-ol	[31]
**4.** **trimeric procyanidin B**	4.2	199, 281	[M−H]^−^	865	695, 577, 407, 289	flavan-3-ol	[32]
**5.** **catechin ^2^**	4.5	232, 278	[M−H]^−^	289	245, 203, 179	flavan-3-ol	[33]
**6.** **dimeric procyanidin B (III)**	5.2	238, 283	[M−H]^−^	577	559, 451, 407, 289	flavan-3-ol	[31]
**7.** **epi-catechin ^2^**	6.6	279	[M−H]^−^	289	245, 203, 179	flavan-3-ol	[32]
**8.** **dihydromyricetin**	7.7	213, 225, 290	[M−H]^−^	319	301, 193, 125	flavonoid	[33]
**9.** **7-hydroxylariciresinol (I) ^3^**	7.8	199, 225, 279	[M−H]^−^	375	357, 345, 327, 297	lignan	[34]
**10.** ** *trans* ** **-astringin**	8.1	192, 218, 324	[M−H]^−^	405	243, 225, 201, 173, 159	stilbene	[33]
**11.** **dihydrokaempferol**	8.4	290	[M−H]^−^	287	259, 243, 181	flavonoid	[34]
**12.** **todolactol ^2^**	9.9	196, 225, 280	[M−H]^−^	375	327, 191, 176	lignan	[33]
**13.** **taxifolin glucoside**	10.2	196, 287	[M−H]^−^	465	447, 437, 303, 285, 259	flavonoid	[35]
**14.** ** *cis* ** **-astringin**	10.8	213, 319	[M−H]^−^	405	243, 225, 201, 173, 159	stilbene	[28]
**15.** **dimeric procyanidin B (IV)**	11.6	243, 280	[M−H]^−^	577	559, 451, 407, 289	flavan-3-ol	[31]
**16.** **7-hydroxylariciresinol (II) ^3^**	11.6	198, 227, 279	[M−H+HCOOH]^−^	421	375, 357, 345, 325	lignan	[34]
**17.** **piceid**	11.7	216, 319	[M−H]^−^	389	227, 185, 183, 157, 143	stilbene	[33]
**18.** **taxifolin ^2^**	12.4	280	[M−H]^−^	303	285, 177	flavonoid	[34]
**19.** **isorhapontin**	12.5	192, 218, 325	[M−H+HCOOH]^−^	465	419, 257, 242, 241	stilbene	[33]
**20.** **cyclolariciresinol ^3^**	12.7	197, 283	[M−H]^−^	359	344, 313, 189	lignan	[34]
**21.** **α-conidendric acid**	14.0	200, 225, 280	[M−H+HCOOH]^−^	419	373, 177	lignan	[34]
**22.** **7-hydroxymatairesinol ^3^**	15.0	197, 226, 280	[M−H]^−^	373	355, 340, 311, 293, 219	lignan	[34]
**23.** **Secoisolariciresinol ^3^**	15.3	232, 281	[M−H]^−^	361	346, 313, 299	lignan	[34]
**24.** **eriodictyol**	15.4	194, 287	[M−H]^−^	287	259, 243, 151	flavonoid	[36]
**25.** **myricetin ^2^**	15.7	254, 373	[M−H]^−^	317	289, 179, 151, 137	flavonoid	[37]
**26.** **secoisolariciresinol guaiacylglyceryl ether**	15.7	197, 280	[M−H]^−^	557	539, 525, 521, 509, 415, 361	sesquilignan	[34]
**27.** **lariciresinol ^3^**	16.1	197, 200, 202, 280	[M−H+HCOOH]^−^	405	359, 329	lignan	[34]
**28.** **lariciresinol guaiacylglyceryl ether**	16.7	201, 225, 280	[M−H]^−^	555	525, 507, 359, 329, 315, 195, 165	sesquilignan	[34]
**29.** **nortrachelogenin ^3^**	18.3	199, 225, 280	[M−H]^−^	373	355, 327, 311, 249, 223, 191, 147	lignan	[34]
**30.** **pinoresinol ^3^**	19.1	201, 280	[M−H]^−^	357	342, 327, 311, 151, 136	lignan	[34]
**31.** **quercetin ^2^**	19.8	208, 368	[M−H]^−^	301	273, 179, 151	flavonoid	[38]
**32.** **matairesinol ^3^**	21.6	197, 281	[M−H]^−^	357	342, 313, 298, 281, 209, 191, 147	lignan	[34]
**33.** **pinobanksin**	22.0	214, 284	[M−H]^−^	271	253	flavonoid	[33]
**34.** **pinosylvin**	26.3	223, 299	[M−H]^−^	211	169	stilbene	[39]
**35.** **pinocembrin**	30.3	214, 288	[M−H]^−^	255	213, 211	flavonoid	[39]
**36.** **pinobanksin 3-*O*-acetate**	30.5	216, 292	[M−H]^−^	313	271, 253	flavonoid	[40]
**37.** **pinosylvin monomethyl ether**	33.6	212, 223, 300	[M−H]^−^	225	210	stilbene	[39]
**38.** **dehydrojuvabione**	41.0	224, 300	[M+H]^+^	265	251, 233, 205, 187, 176	sesquiterpene	[34]
**39.** **neoabietic acid**	52.0	251	[M+H]^+^	303	257, 219, 179, 151, 123	diterpene	[39]
**40.** **abietic acid ^2^**	52.3	241	[M+H]^+^	303	285, 257, 123	diterpene	[39]

^1^ Ionization mode (positive or negative); [M−H]^−^—deprotonated molecule; [M+H]^+^—protonated molecule; [M−H+HCOOH]^−^—formate adduct ion. ^2^ Identified by comparison with reference compounds. ^3^ Identified by comparison with isolated compound characterized by NMR.

**Table 2 cells-11-03332-t002:** Presence of compounds confirmed through LC-DAD–ESI-MS/MS analysis in *Pinaceae* species.

Compounds	*Abies alba*	*Pinus sylvestris*	*Pinus mugo*	*Pinus cembra*	*Pinus strobus*	*Pinus ×rhaetica*	*Larix decidua*	*Larix polonica*	*Larix kaempferi*	*Pseudotsuga menziesii*	*Tsuga canadensis*	*Picea abies*	*Picea glauca*
**1.** **gallocatechin**	+	+	+	+	+	+	+	+	+	−	−	−	−
**2.** **dimeric procyanidin B (I)**	+	+	+	+	+	+	+	+	+	+	+	+	+
**3.** **dimeric procyanidin B (II)**	+	+	+	+	+	+	+	+	+	+	+	+	+
**4.** **trimeric procyanidin B**	+	+	+	+	+	+	+	+	+	+	+	+	+
**5.** **catechin**	+	+	+	+	+	+	+	+	+	+	+	+	+
**6.** **dimeric procyanidin B (III)**	−	−	−	−	−	−	+	+	+	+	−	+	−
**7.** **epi-catechin**	+	−	−	+	+	−	+	+	+	+	+	−	−
**8.** **dihydromyricetin**	−	−	+	−	−	+	+	+	−	−	−	−	−
**9.** **7-hydroxylariciresinol (I)**	+	−	−	−	−	−	−	−	−	−	−	+	+
**10.** ** *trans* ** **-astringin**	−	−	−	−	−	−	−	−	−	−	−	+	+
**11.** **dihydrokaempferol**	−	−	−	−	−	−	+	+	−	−	−	−	−
**12.** **todolactol**	+	−	−	−	−	−	+	−	+	+	+	+	+
**13.** **taxifolin hexoside**	−	+	+	+	+	+	−	+	+	+	−	+	−
**14.** ** *cis* ** **-astringin**	−	−	−	−	−	−	+	+	+	−	−	+	+
**15.** **dimeric procyanidin B (IV)**	−	−	−	−	−	−	+	+	+	−	−	−	−
**16.** **7-hydroxylariciresinol (II)**	+	−	+	−	+	−	+	−	+	+	+	−	−
**17.** **piceid**	−	−	−	−	−	−	−	−	−	−	−	+	+
**18.** **taxifolin**	−	+	+	+	+	+	+	+	+	+	−	+	+
**19.** **isorhapontin**	−	−	−	−	−	−	−	−	−	−	−	+	+
**20.** **cyclolariciresinol**	+	−	−	−	−	−	−	−	−	−	−	−	−
**21.** **α-conidendric acid**	+	−	+	−	+	−	+	+	+	−	+	−	−
**22.** **7-hydroxymatairesinol**	+	−	+	−	+	−	+	+	+	+	+	+	+
**23.** **secoisolariciresinol**	+	−	+	−	+	−	+	+	+	+	+	+	+
**24.** **eriodictyol**	−	−	+	+	+	−	+	+	+	+	−	−	−
**25.** **myricetin**	−	−	+	−	−	−	+	+	+	+	−	−	−
**26.** **secoisolariciresinol guaiacylglyceryl ether**	+	−	+	−	−	−	+	+	+	+	−	−	−
**27.** **lariciresinol**	+	−	−	−	+	−	+	+	+	+	+	−	+
**28.** **lariciresinol guaiacylglyceryl ether**	+	−	−	−	−	−	−	−	−	−	−	−	−
**29.** **nortrachelogenin**	+	+	+	−	+	−	+	+	+	+	+	+	+
**30.** **pinoresinol**	−	+	−	−	−	−	−	−	−	−	−	−	−
**31.** **quercetin**	−	−	−	−	+	−	+	+	+	+	−	−	−
**32.** **matairesinol**	−	+	+	+	+	+	+	+	+	−	+	+	+
**33.** **pinobanksin**	−	−	+	+	+	−	+	+	+	+	−	−	+
**34.** **pinosylvin**	−	+	+	−	+	−	−	−	−	−	−	−	−
**35.** **pinocembrin**	−	+	+	−	+	+	−	−	−	+	−	−	−
**36.** **pinobanksin 3-O-acetate**	−	−	−	−	+	−	−	−	−	−	−	−	−
**37.** **pinosylvin monomethyl ether**	−	+	+	+	+	−	−	−	−	−	−	−	−
**38.** **dehydrojuvabione**	+	−	−	−	−	−	−	−	−	−	−	−	−
**39.** **neoabietic acid**	+	+	+	+	+	+	+	+	+	+	+	+	+
**40.** **abietic acid**	+	+	+	+	+	+	+	+	+	+	+	+	+

## Data Availability

The data presented in this study are available on request from the corresponding author.

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
