# Peer review of "LC-DAD–ESI-MS/MS and NMR Analysis of Conifer Wood Specialized Metabolites"

_cells, 2022, doi:10.3390/cells11203332_

Round 1
Reviewer 1 Report
This paper describes the identification of 42 compounds from a number of trees of the Pinaceae family (13 different species). It appears that the compounds were identified using NMR and LC-MS. Some of the compounds were identified based on comparison to standards. The source of the standards is reported to be compound isolated from this laboratory. The paper tries to suggest that various parts of the pine trees that are harvested by the Polish wood industry could be a source of important compounds. Specifically they report the "knotwood" are not used in the industry. They suggest that this material might have use from the secondary metabolites that have been identified.
The authors could improve this paper if they're able to gther some quantification of the amount of compounds produced by each species of tree. (Added to table 1 or 2.) Perhaps one species produces a significant amount of one metabolite over another.
The identification of some of the compounds is weak at times. In particular when they describe compounds that don't fragment in the ESI spectrum. Perhaps pointing out how it compares with the standards that the lab has claimed to isolate.
The authors use commas when mentioning fractions of percent (i.e, 24,6%. In US English we use periods (i.e., 24.6%). I don't see this as a big issue. Line 423 appears to have an "x" that should not be there.
In the results section, I found their discussion on lignins confusing. I really think this needs to be improved.
I don't think the work is bad. I just question whether there is much interest in this paper. Do the authors really think the metabolites have a use in the chemical industry?
Author Response
The authors could improve this paper if they're able to gther some quantification of the amount of compounds produced by each species of tree. (Added to table 1 or 2.) Perhaps one species produces a significant amount of one metabolite over another.
Table 2 shows the presence of dominating groups of compounds (lignans, stilbenes or flavonoids) in selected species from the Pinaceae family. Together with chromatograms in Supplementary materials this is a way to compare the phytochemical profiles and to select species rich in selected compounds/group of compounds for further chemical and/or biological investigations. Generally, Silver fir wood could be a rich source of lignans, pine wood offer both lignans and stilbenes aglycones, while spruce wood had stilbenes glycosides as dominant metabolites. Larch and Douglas fir wood can be considered abundant sources of flavonoids and flavan-3-ols. But of course, quantitative analysis would be the next step.
The identification of some of the compounds is weak at times. In particular when they describe compounds that don't fragment in the ESI spectrum. Perhaps pointing out how it compares with the standards that the lab has claimed to isolate.
Indeed, we agree with this criticism. In the revised version we are describing more clearly that lignans were isolated during this study and the MS/MS analysis of lignans was based on isolated compounds.
For example :
“Lignans have been identified as major constituents in fir and pine species. However, the spectral data for this class of compound is rather limited, thus for MS analysis of this group of plant metabolites had to be supported by the isolation of studied compounds and elucidation of their structures by NMR. Ten lignans have been isolated using chromatographic methods, namely 7-hydroxylariciresinol (I) (9), todolactol (12), 7-hydroxylariciresinol (II) (16), cyclolariciresinol (20), 7-hydroxymatairesinol (22), secoisolariciresinol (23), lariciresinol (27), nortrachelogenin (29), pinoresinol (30) and matairesinol (32), with their structures confirmed through NMR analysis and comparison with literature data [41,54-56]. Isolated compounds were then used as reference standards in phytochemical profiling of conifer wood extracts.
The 1D NMR and ESI-MS spectra along with NMR chemical shifts of all isolated item are presented in supplementary materials (Figures S98-S107; Table S1).
Based on the LC-DAD-ESI-MS/MS analysis three compounds 9 (tr = 7.8 min), 12 (tr = 9.9min) and 16 (tr = 11.6 min) had their pseudomolecular ion at m/z 375 [M−H]− and UV maxima at around 225 and 280 nm. There have been previously reported three lignan isomers in conifer wood, that could give pseudomolecular ion at m/z 375: liovil, todolactol and 7-hydroxylariciresinol [57]. Unfortunately, no ESI-MS fragmentation pattern data could be found for liovil and 7-hydrolariciresinol., with just one source for todolactol [57]. Through NMR analysis, we were able to assign compounds 9 and 16 as 7-hydroxylariciresinol diastereoisomers, respectively. Fragmentation of m/z 375 ion corresponding to compound 12 gave m/z 327, m/z 191 and m/z 176 which was consistent with fragmentation pattern proposed for todolactol [57], and in agreement with fragmentation of our isolated standard. ….”
The authors use commas when mentioning fractions of percent (i.e, 24,6%. In US English we use periods (i.e., 24.6%). I don't see this as a big issue. Line 423 appears to have an "x" that should not be there.
This was corrected.
In the results section, I found their discussion on lignins confusing. I really think this needs to be improved.
We improved this section.
I don't think the work is bad. I just question whether there is much interest in this paper. Do the authors really think the metabolites have a use in the chemical industry?
It is known from the pharmaceutical and food supplement industry that some preparations based on Pinaceae plant are on the market, such as maritime pine extracts (example Pycnogenol) or 7-hydroxymatairesinol which was patented for ameliorating the metabolic syndrome conditions (EP 3 142 660 B1). Especially, stilbenes and lignans are rather rare and interesting molecules. Lignans and plants rich in these polyphenols class are being reported to possess antimicrobial, anti-inflammatory, hypoglycemic, and cytoprotective activities. Some evidence from clinical and observational studies suggests some dietary lignans (secoisolariciresinol, pinoresinol, lariciresinol, matairesinol, syringaresinol, medioresinol, and sesamin) and their gut microbiota metabolites are associated with reduced risk of some hormone-dependent cancers.
Silver fir wood appeared as a rich source of lignans, pine wood contained both lignans and stilbenes aglycones, while spruce wood had stilbenes glycosides as dominant metabolites. Larch and Douglas fir wood can be considered abundant sources of flavonoids and fla-van-3-ols. So we think that they may be interesting for the pharmaceutical, cosmetic, and/or food industries.
Reviewer 2 Report
1. The title ‘Phytochemical characterization and comparison of Pinaceae species wood occurring in Poland’ is broad. It should reflect the research content of this paper.
2. The manuscript gave a new perspective on the study of many species from the Pinaceae family have been recognized as a rich source of lignans, flavonoids, and other polyphenolics.
However it is not well written.
(1) Please add the identification compounds and their figures using Thin Layer Chromatography.
(2) The conditions of HPLC and UHPLC‐DAD‐ESI‐MS/MS method ( in section 2.7. ) were not in details.
(3) NMR data and spectrograms of the test compounds were missing in this manuscript.
(4) For the same method of expression of ‘UHPLC‐DAD‐ESI‐MS/MS’, it should be unified in the text. For example, ‘LC-MS-MS’ (line 493), etc.
(5) when using an abbreviation first time in this artilcle, please, describe its meaning shortly.
Author Response
- The title ‘Phytochemical characterization and comparison of Pinaceae species wood occurring in Poland’ is broad. It should reflect the research content of this paper.
According to your and Editor’s suggestion, we have changed the title to: “LC-DAD-ESI-MS/MS and NMR analysis of conifer wood specialized metabolites”
- The manuscript gave a new perspective on the study of many species from the Pinaceae family have been recognized as a rich source of lignans, flavonoids, and other polyphenolics.
(1) Please add the identification compounds and their figures using Thin Layer Chromatography.
We used TLC, only for controlling the isolation process. That this why we moved the TLC analysis information to the isolation part to not mislead the readers.
(2) The conditions of HPLC and UHPLC‐DAD‐ESI‐MS/MS method ( in section 2.7. ) were not in details.
We agree. We added more detailed information:
“LC‐DAD‐ESI‐MS/MS analysis was performed on a UHPLC‐3000 RS system (Dionex, Dreieich, Germany) with DAD detection (Dionex, Dreieich, Germany) and an AmaZon SL ion trap mass spectrometer with an ESI interface (Bruker Daltonik GmbH, Bremen, Germany). Separation was performed on a Zorbax SB‐C18 column (150 mm × 2.1 mm, 1.9 μm) (Agilent, Santa Clara, CA, USA). The mobile phase consisted of 0.1% formic acid in water (A) and 0.1% formic acid in acetonitrile (B) using the following gradient: 0–60 min, 15–100% B, then 10 min of equilibration. Samples for LC-DAD-ESI-MS/MS analysis were prepared by dissolving dried extracts in 0.1% formic acid in methanol at the concentration of 10 mg/mL). Standards were prepared in the same way at the concentration of 1 mg/ mL. The flow rate was 0.2 mL/min, Injection injection volume was 5 μL, column temperature was set at 25 °C. The LC eluate was introduced into the ESI interface without splitting, and compounds were analyzed in both positive and negative ion mode with the following settings: nebulizer pressure of 40 psi, drying gas flow rate of 9 L/min; nitrogen gas temperature of 300°C; and a capillary voltage of 4.5 kV. The mass scan ranged from 100 to 2200 m/z….”
(3) NMR data and spectrograms of the test compounds were missing in this manuscript.
All these data: 1H NMR spectra and table with chemical shifts was added to Supplementary Materials
(4) For the same method of expression of ‘UHPLC‐DAD‐ESI‐MS/MS’, it should be unified in the text. For example, ‘LC-MS-MS’ (line 493), etc.
This was corrected
(5) when using an abbreviation first time in this artilcle, please, describe its meaning shortly.
This was done
Reviewer 3 Report
The authors performed good work regarding the chemical data of several Pinaceae species. In my opinion, I do not think that the manuscript is not suitable for Cells readers based on recent Cells papers.
Some major points:
- Chemical structures of compounds should be provided.
- For isolated compounds, their 1D NMR spectra were more important than MS and UV data.
- There are some errors in Table 1. Please check and revise it.
- Regarding the identification part, well-known flavonoids had complete data on MS/MS fragmentation. The authors should cite the relevant refs for each flavonoid class in order to identify them.
- Howevers, lignans, sesquilignans, stibenes, sesquiterpenoids and diterpenoids do not have enough MS/MS data to identify based on your current data and the discussion part. The standards should be provided for HPLC detection. More refs containing data on MS/MS fragmentation of these compounds should be provided. The current data are not convincing.
- I do not understand the Table 2 when the identification of 42 compounds are ambiguous.
- Lacking the conclusion part.
Author Response
The authors performed good work regarding the chemical data of several Pinaceae species. In my opinion, I do not think that the manuscript is not suitable for Cells readers based on recent Cells papers.
Some major points:
- Chemical structures of compounds should be provided.
We have added the structure of all compounds in Supplementary Materials.
- For isolated compounds, their 1D NMR spectra were more important than MS and UV data.
MS, UV data, 1H NMR spectra and a table with chemical shifts were added to Supplementary Materials
- There are some errors in Table 1. Please check and revise it.
This was done, thank you.
- Regarding the identification part, well-known flavonoids had complete data on MS/MS fragmentation. The authors should cite the relevant refs for each flavonoid class in order to identify them.
We have added references for each group of flavonoids identified:
- Razgonova, M.; Zakharenko, A.; Pikula, K.; Manakov, Y.; Ercisli, S.; Derbush, I.; Kislin, E.; Seryodkin, I.; Sabitov, A.; Kalenik, T.; et al. LC-MS/MS Screening of Phenolic Compounds in Wild and Cultivated Grapes Vitis amurensis Rupr. Molecules 2021, 26, 3650.
- Tsimogiannis, D.; Samiotaki, M.; Panayotou, G.; Oreopoulou, V. Characterization of Flavonoid Subgroups and Hydroxy Substitution by HPLC-MS/MS. Molecules 2007, 12, 593-606.
- Koulis, G.A.; Tsagkaris, A.S.; Aalizadeh, R.; Dasenaki, M.E.; Panagopoulou, E.I.; Drivelos, S.; Halagarda, M.; Georgiou, C.A.; Proestos, C.; Thomaidis, N.S. Honey Phenolic Compound Profiling and Authenticity Assessment Using HRMS Targeted and Untargeted Metabolomics. Molecules 2021, 26, 2769.
- Wu, H.; Cao, Y.; Qu, Y.; Li, T.; Wang, J.; Yang, Y.; Zhang, C.; Sun, Y. Integrating UPLC-QE-Orbitrap-MS technology and network pharmacological method to reveal the mechanism of Bailemian capsule to relieve insomnia. Natural Product Research 2022, 36, 2554-2558, doi:10.1080/14786419.2021.1900176.
- Mena, P.; Calani, L.; Dall; #039; Asta, C.; Galaverna, G.; García-Viguera, C.; Bruni, R.; Crozier, A.; Del Rio, D. Rapid and Comprehensive Evaluation of (Poly)phenolic Compounds in Pomegranate (Punica granatum L.) Juice by UHPLC-MSn. Molecules 2012, 17, 14821-14840.
- Howevers, lignans, sesquilignans, stibenes, sesquiterpenoids and diterpenoids do not have enough MS/MS data to identify based on your current data and the discussion part. The standards should be provided for HPLC detection. More refs containing data on MS/MS fragmentation of these compounds should be provided. The current data are not convincing.
We agree with this criticism, in the revised version we are describing more clearly that lignans were isolated during this study, and the MS/MS analysis of lignans was based on isolated compounds. We improved and added citations for MS/MS data for other groups of compounds.
For example :
“Lignans have been identified as major constituents in fir and pine species. However, the spectral data for this class of compound is rather limited, thus for MS analysis of this group of plant metabolites had to be supported by the isolation of studied compounds and elucidation of their structures by NMR. Ten lignans have been isolated using chromatographic methods, namely 7-hydroxylariciresinol (I) (9), todolactol (12), 7-hydroxylariciresinol (II) (16), cyclolariciresinol (20), 7-hydroxymatairesinol (22), secoisolariciresinol (23), lariciresinol (27), nortrachelogenin (29), pinoresinol (30) and matairesinol (32), with their structures confirmed through NMR analysis and comparison with literature data [41,54-56]. Isolated compounds were then used as reference standards in phytochemical profiling of conifer wood extracts.
The 1D NMR and ESI-MS spectra along with NMR chemical shifts of all isolated item are presented in supplementary materials (Figures S98-S107; Table S1).
Based on the LC-DAD-ESI-MS/MS analysis three compounds 9 (tr = 7.8 min), 12 (tr = 9.9min) and 16 (tr = 11.6 min) had their pseudomolecular ion at m/z 375 [M−H]− and UV maxima at around 225 and 280 nm. There have been previously reported three lignan isomers in conifer wood, that could give pseudomolecular ion at m/z 375: liovil, todolactol and 7-hydroxylariciresinol [57]. Unfortunately, no ESI-MS fragmentation pattern data could be found for liovil and 7-hydrolariciresinol., with just one source for todolactol [57]. Through NMR analysis, we were able to assign compounds 9 and 16 as 7-hydroxylariciresinol diastereoisomers, respectively. Fragmentation of m/z 375 ion corresponding to compound 12 gave m/z 327, m/z 191 and m/z 176 which was consistent with fragmentation pattern proposed for todolactol [57], and in agreement with fragmentation of our isolated standard. ….”
For sesquilignans, we were able to find soma MS/MS data and we compared our result.
For example: “Two sesquilignans have been found in conifer wood. Compound 26 (tr = 15.7 min) exhibited a pseudomolecular ion at m/z 557 [M-H]− and fragment ions at m/z 539 [M−H−18]− corresponding to the loss of water, m/z 525 [M−H−32]−, m/z 509 [M−H−48]-]− formed by the loss of formaldehyde and water in the diol structure, m/z 415 and m/z 361 [M−H−196]-]− corresponding to the cleavage of guai-acylglyceryl moiety. The fragmentation pattern was in accordance to the data for oligo-lignans isolated from Norway spruce and Scots pine knots [12]. Based on fragmentation, this compound was tentatively identified as secoisolariciresinol 4-O-guaiacylglyceryl ether…..”
- Willför, S.; Reunanen, M.; Eklund, P.; Sjöholm, R.; Kronberg, L.; Fardim, P.; Pietarinen, S.; Holmbom, B. Oligolignans in Norway spruce and Scots pine knots and Norway spruce stemwood. 2004, 58, 345-354, doi:10.1515/HF.2004.053.
For stiblenes, we were also able to find more MS/MS data and as in the case of 2 compounds (pterostilbene, pinosilvin dimethyl ether), their concentration was to low to obtain reliable MS/MS data, we couldn’t confirm their presence.
For example:
“Compounds 34 (tr = 26.3 min) and 37 (tr = 33.6 min) were characterized as stilbenes based on their primary pseudomolecular ion and UV maxima at around 300 nm. Comparing mass of [M−H]− ion at m/z 211 [M−H−]− and m/z at 225 [M−H]− with literature reports on stilbenes isolated previously from conifer wood and their elution order [53]these compounds were tentatively identified: 34 as pinosylvin and 37 as pinosylvin monomethyl. Moreover, compared to MS/MS data from the literature, pinosylvin (34) demonstrated a fragment ion at 169 m/z [M−H−42]− corresponding to the losses of C2H2O, characteristic for stilbenoids [59,60]. Pinosylvin monomethyl ether (37) pseudomolecular ion gave a fragment ion at 210 m/z [M−H−15]− corresponding to the loss of methyl radical. Their occurrence was limited to the genus Pinus, with only Scots pine and mountain pine contain all the above (Table 2).
- Stella, L.; De Rosso, M.; Panighel, A.; Vedova, A.D.; Flamini, R.; Traldi, P. Collisionally induced fragmentation of [M-H](-) species of resveratrol and piceatannol investigated by deuterium labelling and accurate mass measurements. Rapid Commun Mass Spectrom 2008, 22, 3867-3872, doi:10.1002/rcm.3811.
- Yeo, S.C.M.; Luo, W.; Wu, J.; Ho, P.C.; Lin, H.-S. Quantification of pinosylvin in rat plasma by liquid chromatography–tandem mass spectrometry: Application to a pre-clinical pharmacokinetic study. Journal of Chromatography B 2013, 931, 68-74, doi:https://doi.org/10.1016/j.jchromb.2013.05.023.
We added MS/MS data for dehydrojuvabione and abietic acid.
Manville, J.F. Juvabione and its Analogs. Juvabione and A Dehydrojuvabione Isolated from the Whole Wood of Abies balsamea, have the R,R Stereoconfigurations,not the R,S. Can J Chem 1975, 53, 1579
Schymanski, E. L.; Ruttkies, C.; Krauss, M.; Brouard, C.; Kind, T.; Dührkop, K.; Allen, F.; Vaniya, A.; Verdegem, D.; Böcker, S.; et al. Critical Assessment of Small Molecule Identification 2016: Automated Methods. Journal of Cheminformatics 2017, 9 (1). DOI:10.1186/s13321-017-0207-1
- I do not understand the Table 2 when the identification of 42 compounds are ambiguous.
Table 2 shows the presence of dominating group of compounds (lignans, stilbenes or flavonoids) in selected species from the Pinaceae family. Together with chromatograms in Supplementary materials this is a way to compare the phytochemical profiles and to select species rich in selected compounds/ group of compounds for further chemical and/or biological investigations.
- Lacking the conclusion part.
The conclusion part was added.
Round 2
Reviewer 1 Report
I read the author's comments and feel that these issues have been addressed. Quantification of important metabolites in regards to species would be a good next step for further research. I re-read the paper and agree with the changes that have been made and support the publication of this manuscript.
Author Response
Thank you for your comments.
Reviewer 3 Report
The authors have revised thoroughly the manuscript following the suggestions of the reviewers. In the current version, I do not accept the elucidation of compounds of some stilbenes, sesquilignans, sesquiterpenoids, and diterpenoids which have not MS/MS data in the literature. Compound 28 was said to be lariciresinol 4-O-guaicylglyceryl ether due to the comparison of MS/MS data to those in the literature. I agree this finding when they shared all fragment ions. However, for compound 10, the fragment ion at m/z 243 assigned as piceatannol is ambiguous. How could the authors determine the positions of OH groups? No fragment ions indicated this finding. I suggested that such compounds should be discarded in the elucidation part. Please check and revise those compounds.
Author Response
In the current version, I do not accept the elucidation of compounds of some stilbenes, sesquilignans, sesquiterpenoids, and diterpenoids which have not MS/MS data in the literature.
Thank you for this comment. We have once again revised the manuscript and made sure that there is at least one reference of literature MS/MS data for each compound elucidated there.
Compound 28 was said to be lariciresinol 4-O-guaicylglyceryl ether due to the comparison of MS/MS data to those in the literature. I agree this finding when they shared all fragment ions.
We have once again revised sesquilignans data that we have obtained and added some fragment ions missed before. Both compounds should be now in even better accordance with available literature MS/MS data for sesquilignans.
However, for compound 10, the fragment ion at m/z 243 assigned as piceatannol is ambiguous. How could the authors determine the positions of OH groups? No fragment ions indicated this finding. I suggested that such compounds should be discarded in the elucidation part. Please check and revise those compounds.
Thank you for this comment. Indeed, after further analysis of obtained data, we have added additional fragment ions for these stilbenoids, which should clarify now the structure of aglycones (picaetannol for compounds 10 and 14, and resveratrol for compound 17). We have also added literature data on these fragmentation patterns:
Flamini, R.; De Rosso, M.; De Marchi, F.; Dalla Vedova, A.; Panighel, A.; Gardiman, M.; Maoz, I.; Bavaresco, L. An innovative approach to grape metabolomics: stilbene profiling by suspect screening analysis. Metabolomics 2013, 9, 1243-1253, doi:10.1007/s11306-013-0530-0.
Buiarelli, F.; Coccioli, F.; Jasionowska, R.; Merolle, M.; Terracciano, A. Analysis of some stilbenes in Italian wines by liquid chromatography/tandem mass spectrometry. Rapid Communications in Mass Spectrometry 2007, 21, 2955-2964, doi:https://doi.org/10.1002/rcm.3174.